# Impacts of Land-Use Change on Ecosystem Services Value in the South-to-North Water Diversion Project, China

**DOI:** 10.3390/ijerph20065069

**Published:** 2023-03-13

**Authors:** Jing Zhuge, Jie Zeng, Wanxu Chen, Chi Zhang

**Affiliations:** 1Department of Geography, School of Geography and Information Engineering, China University of Geosciences (Wuhan), Wuhan 430074, China; 2Hubei Key Laboratory of Regional Ecology and Environmental Change, Wuhan 430074, China; 3Key Labs of Law Evaluation of Ministry of Natural Resources of China, Wuhan 430074, China; 4Laboratory of Geospatial Technology for the Middle and Lower Yellow River Regions (Henan University), Ministry of Education, Kaifeng 475004, China; 5Wuhan Geomatics Institute, Wuhan 430022, China

**Keywords:** land-use, ecosystem services value, South-to-North Water Diversion Project, China

## Abstract

The South-to-North Water Diversion Project (SNWD) in China is a trans-basin water transfer project for water resource optimization that affects ecosystem services functions along its main transfer line. Exploring the effects of land-use change on ecosystem services in the headwater and receiving areas along the SNWD is conducive to improving the protection of the surrounding ecological environment. However, previous research lacks a comparative analysis of ecosystem services values (ESVs) in these areas. In this study, the land-use dynamic degree index, land-use transfer matrix, and spatial analysis method were used to comparatively analyze the impact of land-use changes on ESVs in the headwater and receiving areas of the SNWD. The results show that cultivated land was the main land use type in the receiving areas and HAER. From 2000 to 2020, CLUDD in the headwater areas was faster than that in the receiving areas. Spatially, in general, the land-use change areas of the receiving areas were larger. During the study period, cultivated land in the headwater areas of the middle route mainly transferred to water areas and forestry areas, while built-up areas mainly occupied cultivated land in the headwater areas of the east route, receiving areas of the middle route, and receiving areas of the east route. From 2000 to 2020, the ESV increased only in the headwater areas of the middle route, while the ESV in the other three sections decreased. The variation extent of ESV in the receiving areas was greater than that in the headwater areas. The results of this study have important policy significance for land use and ecological protection in the headwater and receiving areas of the SNWD in the future.

## 1. Introduction

The South-to-North Water Diversion Project (SNWD) is the largest inter-basin water transfer project in the world [1], which aims to solve the problem of unevenly distributed water resources in China and promotes sustainable development in water-deficient areas [2,3]. The SNWD includes two diversion project lines, the East Route Project and Middle Route Project, hereafter the east route and the middle route. The SNWD transfers water for industrial and domestic use from areas rich in water resources [4] to areas with water shortages, which affects human activities such as urbanization and economic development [5,6]. These human activities have independently changed the land-use patterns in the headwater and receiving areas of the SNWD, while also causing potential ecological problems and influencing changes in ecosystem service functions (e.g., hydrology regulation, soil conservation, and biodiversity maintenance) [7,8,9]. As a result, the ecosystem services value (ESV) along the SNWD has changed [10,11] (p. 2). Exploring the influence of land-use change (LUC) on ESV can provide a scientific basis for the formulation of land-use and ecological policies along the SNWD line. However, few studies have investigated the differential influence of LUC on ESV in the headwater and receiving areas of the SNWD. Therefore, the temporal-spatial characteristics of the ESV along the SNWD line deserve further investigation.

Ecosystem services are defined as products and services which human beings receive directly or indirectly from the natural ecosystem [12]. ESV quantifies the ecosystem function in monetary form by assigning value, thereby establishing a bridge between the natural ecosystem and the socioeconomic system [13]. It is conducive to the rational allocation of ecological resources to evaluate ESV [14]. Evaluation methods of ESVs are mainly divided into direct methods (i.e., supervisor and objective evaluation) and indirect methods (i.e., energy and material conversion) [15,16]. These evaluation methods are mainly expressed in three forms: money, energy, and physics [17]. Among them, currency evaluation of ESVs links ecosystem services with the market system, which is more convenient for management and policy decisions [18]. Constanza [19] evaluated the value of global ecosystem services and introduced the ESV method into the market system. Compared with previous methods, the ESV method had simple operation characteristics and low data demand, thus it has been widely used [20,21,22]. Xie [23,24] improved the originally proposed ESV method to make it more suitable for the evaluation of ecosystem services in China. Based on the expert knowledge method, services under the classification of ecosystem service functions were revised. This laid a foundation for the evaluation of ESV in China.

LUC refers to the process in that one land type (e.g., cultivated land) is replaced by another land type [25]. LUC has a direct influence on ESV and is the leading factor of ESV change [26]. LUC in terms of structure and quantity often leads to structural changes in ecosystem functions [27], causing several ecological and environmental problems and ultimately affecting the function and value of ecological services [28]. LUC itself is affected by different driving factors. Many studies have shown that LUC is mainly influenced by traffic accessibility, urbanization, economy, and policy [29,30,31,32]. These factors may directly or indirectly affect LUC [33], thereby in turn affecting the changes in ESV. Evaluating the influence of LUC on ESV is helpful to reveal the effect of social and economic activities on ecological service functions and promote sustainable development.

In recent years, changes in the ecological environment along the SNWD line have attracted increasing public attention. As a result, many studies have investigated changes in ESV in the SNWD. For example, in the Three-River Headwaters Basin, a water source area of the SNWD, the ESV in most areas has increased over the past 20 years [34]. The changes in forestry, cultivated, and water areas in the water source area of the middle route caused the ESV to increase [35,36,37]. Whereas, in the SNWD water receiving areas of the Beijing-Tianjin-Hebei region and Shangdong, ESV declined from 2003 to 2008 [38] and 2005 to 2015 [39], respectively. These researches showed that ESV changes in the headwater and receiving areas of the SNWD were different. The reason may be that the redistribution of water resources in the SNWD had different effects on the ecological environment of the headwater and receiving areas. For example, water supply from the water source area to the water receiving area can promote economic and social development in the receiving areas. Previous studies have indicated that the SNWD alleviates water shortage restrictions on urban development and ensures economic development in northern China [40]. The rapid economic and industrial developments have worsened the local environment, thereby affecting the ESV. On the other hand, the SNWD also promoted water conservation in the source areas [41], which alleviated ecological contradictions. These interactions between human activities and the natural environment are directly reflected by LUC [42]. Therefore, by discussing the different influences of LUC on ESV in the headwater and receiving areas along the SNWD line, we can identify more in-depth reasons for the different ESV changes between them. However, previous research has lacked the comparison of temporal-spatial characteristics of ESV in the headwater and receiving areas of the SNWD affected by LUC. Many studies have been conducted on the headwater areas of the SNWD, yet most were concentrated in Hubei and Jiangsu [11,43,44]. Meanwhile, studies on the receiving areas of the SNWD have mainly focused on the Beijing-Tianjin-Hebei region [38,45,46]. Therefore, it is necessary to perform a comparative analysis of the ESV variation trends in all headwater and receiving areas of the SNWD. Further, previous studies are also lacking in terms of a comparative analysis of ESV changes along the east and middle routes of the SNWD, and often selected areas along specific water transmission lines for research. Among them, more studies were conducted on the middle route of the SNWD, while relatively fewer were on the east route [47,48,49,50,51]. This study specifically discusses both the east and middle routes of the SNWD, which has great significance to form protection measures for both ecological environments of the east and middle route regions.

To address the research gap, we used the land-use dynamic degree index and land-use transfer matrix to explore the dynamics of LUC in the headwater and receiving areas of the SNWD. The value equivalent method was used to evaluate ESV in the headwater and receiving areas respectively, and the spatial and temporal characteristics of ESV were explored by GIS spatial analysis. This study was mainly aimed at addressing the following questions: (1) How did land use in the headwater and receiving areas of the SNWD respectively change in terms of quantity and space from 2000 to 2020? (2) What are the differences in ESV change between the headwater and receiving areas from 2000 to 2020? (3) What is driving the impact of the SNWD on ESV in its headwater and receiving areas? The results of this study provide a scientific basis for land-use and ecological management planning in distinct areas along the SNWD line.

## 2. Study Area

The SNWD (Figure 1) spans the eastern and middle regions of China (108°35′40″ E–121°27′47″ E, 30°40′11″ N–40°11′8″ N). The main climate types of the study area are subtropical monsoon and temperate monsoon, with mean annual precipitation ranging from 379.5–1772.2 mm. The main terrain of the study area is plains; however, mountainous regions and plateaus cover a small area. The east route diverts water from Yangzhou to Shandong and Tianjin northwards, and the middle route diverts water from the Danjiang River to Beijing. According to the existing literature [52,53,54,55], the SNWD is divided into the headwater (HAER) and receiving (RAER) areas of the east route, and the headwater (HAMR) and receiving (RAMR) areas of the middle route. Geographically, the HAER covers Shanghai, Zhejiang, and southern Jiangsu, while the RAER covers Shandong, eastern Hebei, eastern Tianjin, and northern Jiangsu (see Appendix A). The HAMR covers northwestern Hubei, southeastern Shaanxi, southeastern Chongqing, and southern Henan, while the RAMR covers Beijing, western Tianjin, northern Henan, and western Hebei. By 2017, up to 29.86 billion m^3^ of water resources were being diverted from the HAER [56]. By 2018, more than 190 billion m^3^ of water resources were being diverted from the HAMR [57]. The redistribution of water resources promoted ecological circulation, improved industry and agricultural production conditions, and promoted the coordinated development of the economy and society in the receiving areas [40]. Meanwhile, pollution in the HAER was found to be of greater severity than that in the RAER [58]. Overall, it has been determined that inter-basin water diversion projects in China have increased difficulties in pollution control and affected ecological benefits, thereby ultimately influencing the sustainable development of the economy and society.

## 3. Materials and Methods

### 3.1. Data Sources

Land-use and normalized difference vegetation index (NDVI) data were obtained from the Data Center for Resources and Environmental Sciences at the Chinese Academy of Sciences (http://www.resdc.cn, accessed on 2 March 2021). Based on Landsat 8 remote sensing image data, land-use data (2000, 2005, 2010, 2015, and 2020) were generated at a 1 km resolution by manual visual interpretation. According to the Classification Standard of the Chinese Ecosystem, Land Use Status Classification, and the land-use classification of the image data, land-use data in this study were divided into seven types: cultivated land, forestry areas, grassland, water areas, built-up areas, unused land, and wetland. The comprehensively identified accuracy of the land-use data sets for all seven land-use types was over 95% [59]. Grain and economic data were obtained from the China Statistical Yearbook (2008), National Farm Product Cost-benefit Survey (2008), and China Agriculture Statistical Report (2008). Data on total water resources from 2005 to 2019 were obtained from the National Bureau of Statistics (http://www.stats.gov.cn/, accessed on 30 June 2021), the National Water Resources Bulletin (2000), the Jiangsu Statistical Yearbooks (2001–2020), the Henan Water Resources Bulletins (2000–2019), and the Hebei Water Resources Bulletin (2000). Data for 2020 were missing; hence data for 2020 were substituted by those of 2019.

### 3.2. Research Methods

#### 3.2.1. Land-Use Dynamic Degree Index

The land-use dynamic degree index, which is comprised of a single land-use dynamic degree index (SLUDD) and a comprehensive land-use dynamic degree index (CLUDD), was used for indicating the rates of LUC [60]. SLUDD refers to the quantity change of a certain type of land at one point, reflecting the change in the degree of a certain land-use type. CLUDD reflects the comprehensive changes in all land-use types. Calculation equations of SLUDD and CLUDD are shown below:(1)$$SLUDD=\frac{A_{n}-A_{m}}{A_{m}}\times \frac{1}{T}\times 100\%$$
(2)$$CLUDD=\left[\frac{\sum_{i=1}^{n}\Delta LU_{i-j}}{\sum_{i=1}^{n}LU_{i}}\right]\times \frac{1}{T}\times 100\%$$
where *A_m_* represents the measure of single land-use at the initial period, *A_n_* represents the measure of single land-use at the final period, $\Delta LU_{i-j}$ is the area of land-use type *i* changes to land-use type *j* (*j* = 1, 2,…*n*, *i* ≠ *j*), $LU_{i}$ represents the measure of land-use type *i* at the initial period, and *T* represents the study period.

#### 3.2.2. Land-Use Transfer Matrix

The land-use transfer matrix can clearly reflect the direction and quantity of change of all land-use types and is suitable for describing the dynamic change in land use [61]. This can be calculated using Equation (3):(3)$$S_{ij}=\left[\begin{array}{cccc} S_{11} & S_{12} & \ldots & S_{1n}\\ S_{21} & S_{22} & \ldots & S_{2n}\\ \ldots & \ldots & \ldots & \ldots \\ S_{n1} & S_{n2} & S_{n3} & S_{nn} \end{array}\right]$$
where *S_ij_* represents areas of all land types, *n* represents the number of land-use types, *i* denotes the types of land use in the initial stage of the study, and *j* denotes the types of land use in the final stage of the study.

#### 3.2.3. Calculation of the ESV

We used the equivalence factor method for ESV evaluation, which has the advantage of a wide range of applications and fewer variable requirements [19,23]. It can be calculated using Equation (4):(4)$$ESV=\sum_{i=1}^{n}\sum_{j=1}^{m}A_{i}\times VC_{ij}$$
where *A_i_* represents type *i* of land use, *VC_ij_* represents the *j*th type of the ESV coefficient (VC) in land use type *i*, *n* represents the number of land use types, and *m* represents the quantity of ecosystem services types.

The ESV equivalent per unit area of China in 2007 was adopted for calculations in this study [24]. According to previous research [62], the built-up area can be set to 0. Due to the grain output and unit price difference between locations, we revised the value of the equivalent from national to regional where the provinces and municipalities directly under the central government are located in terms of price and output, respectively. Based on this, the economic value of grain according to data from 11 provinces and municipalities was calculated. The data used in this study were based on grain prices from 2007 to replace that of each year in the study period. Since cereals produced under natural conditions are lesser than those produced under the present conditions of human intervention, the economic value of the equivalent can be calculated using one-seventh of the product of the average grain output and unit price [7]. The equation is as follows:(5)$$DK=\frac{1}{7}\times \frac{\sum q_{ki}\times p_{ki}\times m_{ki}}{\sum m_{ki}}$$
where *DK* represents the economic value of grain provided per unit area in region *k*, *q_ki_* is the average unit area yield of cereal in land-type *i* in region *k*, *p_ki_* is the average unit price of cereal in land-type *i* in region *k*, and *m_ki_* is the sown area of cereal in land-type *i* in region *k*.

As shown in Table 1, the economic value of grain provided per unit area for each province/municipality directly under the central government is presented.

The ESV can be adjusted based on biomass [24]. The biomass differs between different regions, and the NDVI has a positive relationship with biomass; the larger the biomass, the larger the ESV [63]. Vegetation coverage can be calculated using NDVI. Therefore, NDVI can be used to calculate the equivalent value. In this study, the equivalent value is revised to the grid. The specific formulas are shown in Equations (6)–(8) [64]:(6)$$f=\frac{NDVI-NDVI_{\min}}{NDVI_{\max}-NDVI_{\min}}$$
(7)$$f_{vi}=\frac{f_{i}}{\bar{f_{k}}}$$
(8)$$E_{f_{vi}}=E_{i}\times f_{vi}$$
where *f* is vegetation coverage, *f_vi_* represents the revision factor of grid *i*, *f_i_* represents the vegetation coverage of grid *i*, ?*f_k_* is the average vegetation coverage of province/municipality directly under the central government *k*, $E_{f_{vi}}$ is the ESV equivalent revised of grid *i*, and *E_i_* is the ESV equivalent before grid *i* revision.

#### 3.2.4. Change Rate of ESV

The elastic coefficient is commonly used to measure the increase in the rate of one or several variables. To measure the change direction and speed of ESV of all partitions in all research periods, we used the elastic coefficient as a measure index based on previous research results [65]. It can be calculated using Equation (9):(9)$$C_{i}=\frac{ESV_{end}-ESV_{start}}{ESV_{start}}\times 100\%$$
where *C_i_* is the elastic coefficient of ESV, *ESV_start_* is the ESV at the initial period, and *ESV_end_* is the ESV at the final period.

#### 3.2.5. Sensitivity Index

The sensitivity index can measure the degree of ESV and is dependent on the ESV coefficient, which is suitable for testing the effectiveness of ESV. The elastic coefficient is used to calculate the sensitivity index. The principle is that the determined ESVs of all types of land use are changed by 50% [32], respectively. This can be calculated by Equation (10):(10)$$CS=\left|\frac{\frac{ESV_{j}-ESV_{i}}{ESV_{i}}}{\frac{VC_{j}-VC_{i}}{VC_{i}}}\right|$$
where *CS* is the sensitivity index, *ESV_i_* is the ESV before adjustment and *VC_i_* is the ESV coefficient before adjustment, respectively; *ESV_j_* is the ESV after adjustment and *VC_j_* is the ESV coefficient after adjustment, respectively.

According to the previous literature [66], *CS* is always less than 1. The larger the *CS* is, the greater the sensitivity of the study region is, and the accuracy of the equivalent coefficient is more critical.

## 4. Results

### 4.1. Land-Use Change before and after Water Supply

The results showed that cultivated land was the main land type in the HAER, RAER and RAMR, which had been decreasing during the study period, mainly being transformed into construction land. The CULDD in headwater areas was greater than in receiving areas. Generally speaking, spatially, land use transformation occurred more in receiving areas. Among them, the transformation of various land use types in the RAER was more intense. Classified according to ecosystem service functions, the ESVs of all ecosystem service functions in the HAER and receiving areas decreased during the study period.

#### 4.1.1. LUCs in the HAMR

From 2000 to 2020, the main land use types in the HAMR were forestry areas, cultivated land and grassland, which accounted for over 44%, 29% and 20% respectively in 2000, 2005, 2010, 2015, and 2020 (Appendix A). During the study period, built-up, water, and wetland areas changed the fastest (Figure 2), respectively, with the SLUDD of 2.50%, 1.55%, and −1.17%, respectively. Wetland decreased the most rapidly, with the SLUDD of −1.17%. Water areas and built-up areas continued to grow, while cultivated land gradually decreased. CLUDD of HAMR was 1.81% from 2000 to 2020. The CLUDD across the four periods first showed a decreasing trend, then an increasing one. 

The land-use transformations in the HAMR are shown in Appendix A. From 2000 to 2005, water areas became the largest transfer into land use type, with an area of 408.17 km^2^, mainly from cultivated land and wetland. Cultivated land was the largest transfer-out land use type, with an area of 607.24 km^2^, mainly into water, grassland, and built-up areas. From 2005 to 2010, the largest transfer-into land use types were built-up and water areas, with areas of 102.44 km^2^ and 84.62 km^2^, respectively. Their main source was cultivated land. Cultivated land was the largest transferred-out land use type, with an area of 181.59 km^2^. It was mainly transferred into built-up, water, and forestry areas. From 2010 to 2015, cultivated and forestry areas were mainly transferred into each other. The transfer-out areas were 7668.25 km^2^ and 6613.14 km^2^, respectively, and the transfer-into areas were 7376.11 km^2^ and 6299.02 km^2^, respectively. From 2015 to 2020, the largest transfer-into land use type was forestry areas, with an area of 17,556.98 km^2^, mainly from grassland and cultivated land. Cultivated land was the most transfer-out land use type, with an area of 17,318.02 km^2^. It was mainly transferred into grassland and forestry areas.

#### 4.1.2. LUCs in the HAER

Cultivated land was the largest land-use type in the HAER (Appendix A), accounting for more than 50% during the study period. In addition, it decreased the fastest among all reduced land types from 2000 to 2020 (Figure 2), with the SLUDD reaching −1.01%. Unused land and built-up areas increased the fastest from 2000 to 2020, with the SLUDD of 11.75% and 6.22%, respectively. Grassland, water, and built-up areas increased gradually, while cultivated land decreased gradually. The CLUDD of the HAER was 1.70% from 2000 to 2020. 

The land-use transfers in HAER are shown in Appendix A. From 2000 to 2005, the largest transfer into land use type was built-up areas, with an area of 1780.70 km^2^. Its main source was cultivated land. Cultivated land was the most transfer-out land use type, with an area of 2048.23 km^2^. It was mainly transferred into built-up areas and water areas. From 2005 to 2010, the largest transfer into land use type was built-up areas, with an area of 1540.59 km^2^. Its main source was cultivated land. Cultivated land was the most transfer-out land use type, with an area of 1578.62 km^2^. It was mainly transferred into built-up areas. From 2010 to 2020, it was mainly that cultivated land and built-up areas were transferred into each other. 

#### 4.1.3. LUCs in the RAMR

During the study period, cultivated land still became the main land type in the RAMR (Appendix A). As shown in Figure 2, the trend of single-type LUC in four periods was similar to that in the HAER. Unused land, built-up areas, and water areas changed the fastest from 2000 to 2020, with the SLUDD of −2.99%, 2.93%, and 2.00%, respectively. Among them, unused land decreased the fastest and built-up areas increased the fastest. CLUDD of the RAMR was 1.50% from 2000 to 2020. The CLUDD of the RAMR decreased first and then increased in the four time periods. 

During the study period, all land types in the RAMR were transferred into each other (Appendix A). From 2000 to 2005, it was mainly that cultivated land and built-up areas were transformed into each other. Cultivated land was transferred out 2660.17 km^2^, which was the main transfer-out land use type. Built-up areas and cultivated land were the main transfer-into land use types, with areas of 2161.29 km^2^ and 1230.52 km^2^, respectively. From 2005 to 2010, the transformation between land types was mainly from cultivated land into built-up areas, with a transfer-out area of 815.24 km^2^ and a transfer-into area of 751.00 km^2^. From 2010 to 2020, the transformation trend of various land types was similar to that from 2000 to 2005. 

#### 4.1.4. LUCs in the RAER

As shown in Appendix A, cultivated land was the largest land-use type in the RAER, with area ratios all over 67% in 2000, 2005, 2010, 2015, and 2020. Only built-up areas grew over the past two decades, while cultivated land decreased across the study period (Figure 2). Unused land and water areas changed the fastest from 2000 to 2020, with the SLUDD of −4.33% and 2.94%, respectively. The CLUDD of the RAER was 0.50% from 2000 to 2020, exhibiting the smallest value among the four regions.

From 2000 to 2005, the largest transfer-into areas were cultivated land and built-up areas, with areas of 16,646.97 km^2^ and 11,685.93 km^2^ (Appendix A), respectively. Cultivated land was mainly transferred from built-up areas, grassland, and forestry areas, while the main source of built-up area transfer was cultivated land. Cultivated land transferred out the most, with an area of 17,941.02 km^2^, mainly into built-up areas and grassland. From 2005 to 2015, cultivated land was mainly transferred into built-up areas. During 2005–2010 and 2010–2015, transferred-into areas of built-up areas were 1733.49 km^2^ and 1179.66 km^2^, respectively, while transferred-out areas of cultivated land were 1619.21 km^2^ and 1201.73 km^2^, respectively. From 2015 to 2020, in all land types, it was mainly cultivated land and built-up areas were transformed into each other. Cultivated land was transferred out of 27,253.57 km^2^ and transferred into 25,221.44 km^2^. Built-up areas were transferred out of 17,707.28 km^2^ and transferred into 20,969.92 km^2^.

#### 4.1.5. Spatial Characteristics of LUC

The LUCs in all regions of the SNWD are shown in Figure 3. The changes were mainly from wetlands, cultivated land, and grasslands to other land types. In most periods, LUCs were mainly concentrated in the receiving areas, and the transformation between land-use types was more intense in the RAER than in the RAMR. The proportions of water areas, wetlands, and cultivated land changed into other land types were largest from 2000 to 2005. The variable land use was mainly distributed in the RAER. The extent of wetland and cultivated land transformed into other land types was large from 2005 to 2010. They were mainly distributed in the HAER. The areas of wetland, cultivated land, and grassland transferred to other land types were the largest from 2010 to 2015 and were mainly distributed in the headwater areas. The changes in cultivated land and grassland into other land types were large from 2015 to 2020.

### 4.2. Changes in ESVs before and after Water Supply

We calculated the changes in the ESVs of the four regions using the equivalent factor method. From 2000 to 2020, the ESVs increased in the HAMR and decreased in the other three subareas. The variation extent of the ESVs in the receiving areas was greater than in the headwater areas. In space, the change rate of the ESVs decreased from the centre to the periphery, and the change rates of the ESVs of the receiving areas were higher than those of the headwater areas. According to the land-use types across the whole study area, the ESV of cultivated land decreased the most and that of water areas increased the most from 2000 to 2020.

#### 4.2.1. Changes in ESVs in the HAMR

The ESV of forestry areas was the highest of all land-use types in 2000, 2005, 2010, 2015, and 2020, occupying over 73% of the total value (Table 2). The reason for this was that forestry areas were the largest land type in this region (Appendix A). The ESVs of the other land-use types in the HAMR ranked from highest to lowest as follows: cultivated land > grassland > water areas > wetlands > unused land. During the study period, the ESV of the HAMR increased by 870.71 million yuan. From 2000 to 2020, the land type with an ESV of the greatest decrease was cultivated land, which decreased by 457.45 million yuan. Water areas increased the most, with a value of 1401.35 million yuan. Across the four periods, the ESV of cultivated land had been decreasing, while that of water areas had gradually increased.

Sorted by ecosystem service functions, the ESVs in the HAMR from high to low were as follows: water regulation > biodiversity maintenance > soil conservation > climate regulation > gas regulation > hydrology regulation > raw material production > aesthetic landscape provision > and food production (Appendix A). Combined with changes in ecosystem service functions during the four-time intervals, only the values of food production had been decreasing, whereas that of hydrology regulation and waste treatment gradually increased. During 2000–2020, the value of hydrology regulation increased the most, reaching 485.33 million yuan, and the value of soil conservation decreased the most, reaching 49.39 million yuan.

#### 4.2.2. Changes in ESVs in the HAER

According to the land use types, the ESVs in the HAER ranked from high to low as follows: cultivated land > water areas > forestry areas > wetland > grassland > unused land (Table 3). The ESV in the HAER decreased by 2217.80 million yuan across the study period. The decrease in the ESV of the HAER gradually increased from 2000 to 2020. There was a constant decrease in the ESV of cultivated land from 2000 to 2020. From 2000 to 2020, among the seven land use types, the ESV of cultivated land decreased the most, by 2337.08 million yuan. In turn, the ESV of water areas increased the most, by 349.44 million yuan during 2000–2020. 

Sorted by ecosystem service functions, the ESVs in the HAER from high to low were as follows: water regulation > hydrology regulation > soil conservation > biodiversity maintenance > climate regulation > gas regulation > food production > aesthetic landscape provision > raw material production (Appendix A). From 2000 to 2020, the values of soil conservation, hydrological regulation, and climate regulation decreased the most, by 440.08 million, 355.08 million, and 327.72 million yuan, respectively. The ESVs of food production, raw material production, gas regulation, climate regulation, soil conservation, and biodiversity maintenance had been decreasing from 2000 to 2020.

#### 4.2.3. Changes in ESVs in the RAMR

The ESV of cultivated land was the highest among all land types during the study period, occupying over 48% of the total value (Table 4). The ESV of other land-use types ranked from high to low are as follows: forestry areas > grassland > water areas > wetland > unused land. The ESV of the RAMR decreased by 2744.98 million yuan from 2000 to 2020. During the study period, the land-use type whose ESV decreased the most was cultivated land, decreasing by 3220.80 million yuan. The land-use type whose ESV increased the most was water areas, which increased by 1799.44 million yuan. During the four time periods, only the ESV of cultivated land had been decreasing.

Sorted by ecosystem service functions, the ESVs in the RAMR from high to low were as follows: soil conservation > hydrology regulation > water regulation > biodiversity maintenance > climate regulation > gas regulation > food production > raw material production > aesthetic landscape provision (Appendix A). Only the value of hydrology regulation increased during the study period, by 101.23 million yuan. The ESVs of food production, raw material production, gas regulation, soil conservation, and biodiversity maintenance had been decreasing from 2000 to 2020. During the study period, the values of climate regulation and soil conservation decreased the most, reaching 647.52 million and 634.68 million yuan, respectively.

#### 4.2.4. Changes in ESVs in the RAER

According to the land use types, the ESVs in RAER ranked from high to low as follows: cultivated land > water areas > forestry areas > wetland > grassland > and unused land (Table 5). The ESV in the RAER increased by 3160.31 million yuan during the study period. The ESV of cultivated land decreased the most from 2000 to 2020, by 2089.90 million yuan. The only land use type with an increasing ESV was water areas, which increased by 3251.90 million yuan during 2000–2020. During the study period, the ESVs of cultivated land and unused land had been decreasing, while that of water areas had been increasing.

Sorted by ecosystem service functions, the ESVs in the RAER ranked from high to low as follows: hydrological regulation > water regulation > soil conservation > biodiversity maintenance > climate regulation > gas regulation > food production > raw material production > aesthetic landscape provision (Appendix A). Except for hydrology regulation, waste treatment, and aesthetic landscape provision, the ESVs of other ecosystem service functions in the RAER had been decreasing from 2000 to 2020. The value of hydrology regulation showed the greatest increase, reaching 431.86 million yuan. The values of soil conservation and climate regulation decreased the most, reaching 942.77 million and 830.35 million yuan, respectively.

#### 4.2.5. Sensitivity Index

As shown in Figure 4, the sensitivity indices of the ESVs in the four regions of the middle route were all less than 0.75 for all land types, with most sensitivity indices being less than 0.3. This means that the ESVs of all land types were independent of the VC and that the ESV had little dependence on the VC. The ESVs of all land-use types in the four regions of the middle route passed the sensitivity index test.

#### 4.2.6. Spatial Characteristics of ESV Changes

The change rate of the ESVs reflected a circular structure that spatially decreased from the centre to the periphery (Figure 5). The change rates of land use in the study area at all intervals were mainly between −14.76% and 5.5%. Elasticity indices of ESVs between −100.00% and −14.77% were mainly distributed in the receiving areas and the HAER. The elasticity index of ESVs from 5.5% to 155.24% was mainly distributed in the RAER. It can be seen that four subareas ranked from high to low according to the change rates of ESV are as follows: RAER > HAER > RAMR > HAMR. Across the four time periods, areas with an ESV elasticity index in the range of −100.00–−14.77% and 5.5–155.24% were most widely distributed during 2015–2020. This meant that the ESVs changed most frequently between 2015 and 2020.

## 5. Discussion

### 5.1. Difference in the Impacts of LUCs on ESVs in the SNWD

Different land-use patterns can be expected to lead to spatial differentiation characteristics of ESV. From 2000 to 2020, the ESV in the HAMR generally increased. This is consistent with previous research results [67,68]. The water area was the most transferred land type from 2000 to 2010 and forestry land was the most transferred land type from 2010 to 2020 in the HAMR, which led to the increase of ESV in the HAMR. Moreover, the SNWD enhanced the water storage functionality of reservoirs, thereby promoting water conservation in the HAMR [41]. This is also supported by the results showing a continuous increase in total water resources in the HAMR from 2000 to 2010 (Figure 6). Therefore, the ESV of the water regulation function was greatly increased. The expansion of forestry land led to the increase of ESV during 2010–2020. The increase in forest land may be due to the policy of returning farmland to the forest [69]. During the study period, the ESV in the HAER decreased. This result is consistent with the research of Meng [37] and Fu [70], thereby indicating that our analysis is reliable. Although the water area was the most transferred land type from 2000 to 2005 in the HAER, the built-up area was the most transferred land type from 2005 to 2020, which directly led to the reduction of ESV in the HAER. The SNWD was beneficial to water conservation in the HAER, and the total amount of water resources gradually increased from 2000 to 2015 (Figure 6), which promoted the increase of ESV to some extent. However, with the development of urbanization, the built-up areas in the HAER expanded rapidly. The ESV of other land types occupied by the built-up areas was higher, causing the overall ESV of the HAER to decrease. In a study of Jiangsu Province, Gao [10] showed that urbanization was a key factor of ESV change in the HAER. The reason for the different changes of ESV in the HAER and HAMR was the faster urbanization in the HAER, which led to the large-scale expansion of built-up areas. The rapid expansion of built-up areas reduced the ESV in the HAER. Among all land types, built-up areas occupied the largest area of cultivated land in the HAER. Therefore, it is necessary to balance urban development and ecological protection, specially cultivated land protection, in the HAER.

From 2000 to 2020, ESV in the receiving areas decreased, and changed to a greater extent than in the headwater areas. Previous studies have also reached similar conclusions [71,72]. From 2000 to 2020, of all land types, built-up areas were transferred in the most, and cultivated land was transferred out the most. This shows that during the study period, the built-up area in the receiving areas was constantly expanding, while the cultivated land was occupied in large quantities, which caused the ESV of receiving areas to decline. As the SNWD transfers water from the headwater areas to provide industrial, agricultural, and domestic water for the receiving areas, the urban water pressure in the receiving areas is alleviated, while urban development potential is increased and the rate of economic development is high. Urban and industrial development has increased the built-up areas and occupied a large amount of previously cultivated land. This is the main reason for the change of ESV in the receiving areas. Hubei and Henan are important grain-producing areas, and the Guanzhong Basin is the grain base of Shaanxi [73,74]. To protect food security, the change rate in cultivated land had been relatively slow and the occupied cultivated lands were relatively small. Therefore, urbanization development of headwater areas, especially those along the middle route, lagged behind, and the ecological environments maintained a higher quality. Therefore, the ESV of the receiving areas decreased at a greater rate from 2000 to 2020. However, the total amount of water resources in the receiving areas generally declined from 2000 to 2020 (Figure 6). This shows that SNWD had not yet met the water demand of the receiving areas by supplementing water resources, which lowered the value in terms of ecological benefit for the receiving area. From 2000 to 2020, the decline of ESV in the RAER was greater than that in the RAMR. We also found that compared with the RAMR, the transformation degree of different land types in RAER was higher. This shows that drastic LUC was a possible reason which led to more obvious ecological degradation in the RAER. Previous studies have also reached similar conclusions [38,72]. Therefore, it is necessary to rationally plan the land-use pattern of the receiving areas, and improve its water conservation function on the basis of maintaining economic development.

### 5.2. Policy Implications

As a sustainable water resource management policy, the SNWD has a decisive impact on changing the ecosystems of headwater and receiving areas [75]. For example, the government regulates the allocation of water resources by adjusting water prices, line operations, and fiscal expenditures [76,77]. To some extent, these policy measures can relieve the ecological and environmental pressure in the receiving areas. However, inter-basin water transfer has promoted the economic development of the receiving areas, which may in turn destroy the ecosystem of those receiving areas [75]. Therefore, the decline of ESV in the receiving areas was within reasonable expectation. In addition, the huge difference in water supply between the east and middle routes of the SNWD has caused different changes in ESV in the areas along these two water transmission lines [78]. The water transport capacity of the middle route is much larger than that of the east route, which relatively slows down the downward trend of ESV in the RAMR. The HAMR provides a greater amount of water resources. The local water storage and water conservation capacity was significantly improved, and ESV increased. Sustainable development is an important strategy to realize the benign development of regional ecology and economy, and improving the ecological service function is an important content to realize this strategy. Measures such as ecological compensation, government management, and ecological management can improve ecosystem services.

For this reason, we put forward the following suggestions for the headwater and receiving areas of the two lines respectively:(1)Establish a reasonable ecological compensation mechanism between the headwater and receiving areas. The receiving areas should pay the economic losses in the process of water supply in the headwater areas, which is used to coordinate the contradiction between economic development in the headwater areas and ensuring water supply [10]. Among them, the ESV in the HAER was decreasing continuously throughout the study area; therefore, more compensation is needed. However, due to water shortages in the receiving areas, ESV showed a downward trend. Thus, the compensation standard of receiving areas should also consider the local economic development.(2)In the management of land resources, the rational use of land should be strengthened. The disorderly expansion of urban construction land should be strictly controlled along the east route and the RAMR to relieve the deterioration of the ecological environment caused by rapid urbanization.(3)Except for the HAMR, the ESV of the other three areas decreased from 2000 to 2020. Therefore, the ecological management of the RAMR, RAER, and HAER should be strengthened to alleviate the degradation of ecosystem functions in these areas. In particular, the receiving areas should become the focus of ecological protection and ecological governance.

### 5.3. Limitations and Future Directions

The land-use dynamic degree index, land-use transfer matrix, and spatial statistical method were used to explore the change law of LUC and ESV along the SNWD, and we found different changes of ESV in the headwater and receiving areas of the SNWD for the duration of the study period. However, this study focused on the economic value of ecosystem services, which is characterized by the total amount of money. This method ignores some internal information such as society and ecology, as well as the influence of human will on ESV. In the future, ESV calculations should be extended to the social and public levels.

## 6. Conclusions

In this study, we chose the middle and east routes of the SNWD as the research area, and separately compared the impacts of LUC in headwater and receiving areas on the ESVs in two routes, respectively. The main conclusions of the evaluation are:(1)From 2000 to 2020, the main land-use type in the receiving areas and the HAER was cultivated land, while the main land-use types of the HAMR were forestry areas, cultivated land, and grassland. The CLUDD value of the water source area was greater than that of the receiving areas.(2)From the perspective of LUC transformation, all land-use types were frequently transformed. Cultivated land in the HAMR was mainly transformed into water and forestry land, while cultivated land in other regions was mainly transformed into built-up areas. From a spatial perspective, the extent of LUCs in the receiving areas was larger than in the headwater areas.(3)From 2000 to 2020, the ESVs increased only in the HAMR. The variation extent of the ESVs of the receiving areas was greater than that of the headwater areas. Spatially, the ESV variation rate in the receiving areas was higher than that in the headwater areas.(4)The ESVs decrease in the HAER and receiving areas may be due to the expansion of construction land caused by local economic development, which damaged the ecosystem function. It was difficult for the water supply to meet the water resources demand of the receiving areas and, thus, the reduction in the ESVs of the receiving areas was greater. The difference in ESVs between the middle and the eastern line may be caused by differences in the water supply.

## Figures and Tables

**Figure 1 ijerph-20-05069-f001:**
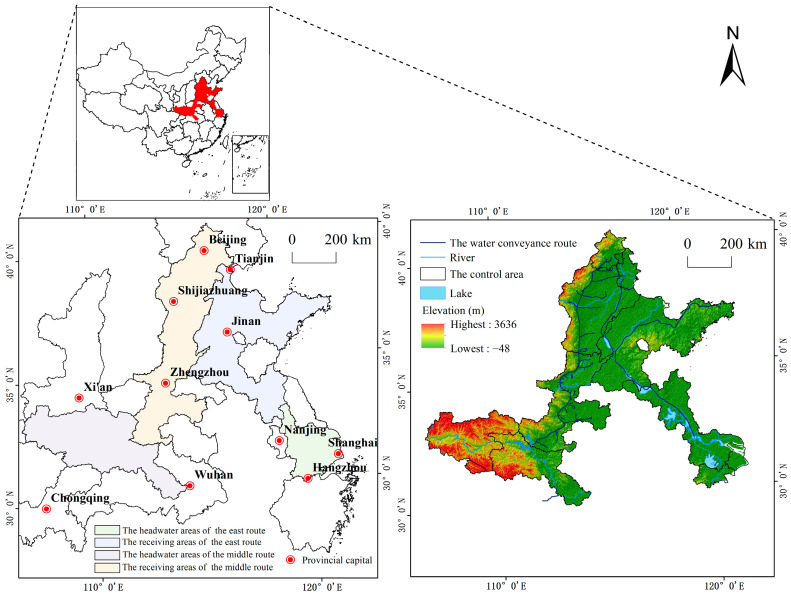
Location of the study area.

**Figure 2 ijerph-20-05069-f002:**
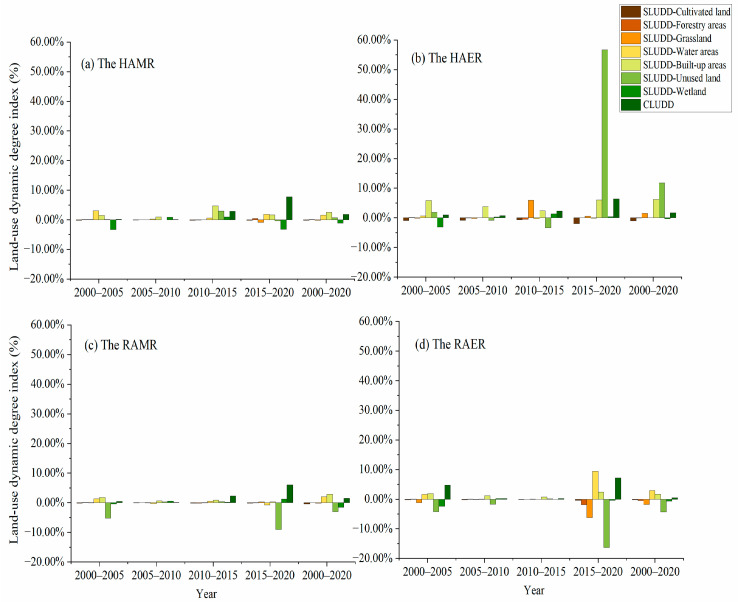
Land use dynamic degree of the HAMR, HAER, RAMR, and RAER from 2000 to 2020. (**a**) Land use dynamic degree of the HAMR, (**b**) Land use dynamic degree of the HAER, (**c**) Land use dynamic degree of the RAMR, (**d**) Land use dynamic degree of the RAER.

**Figure 3 ijerph-20-05069-f003:**
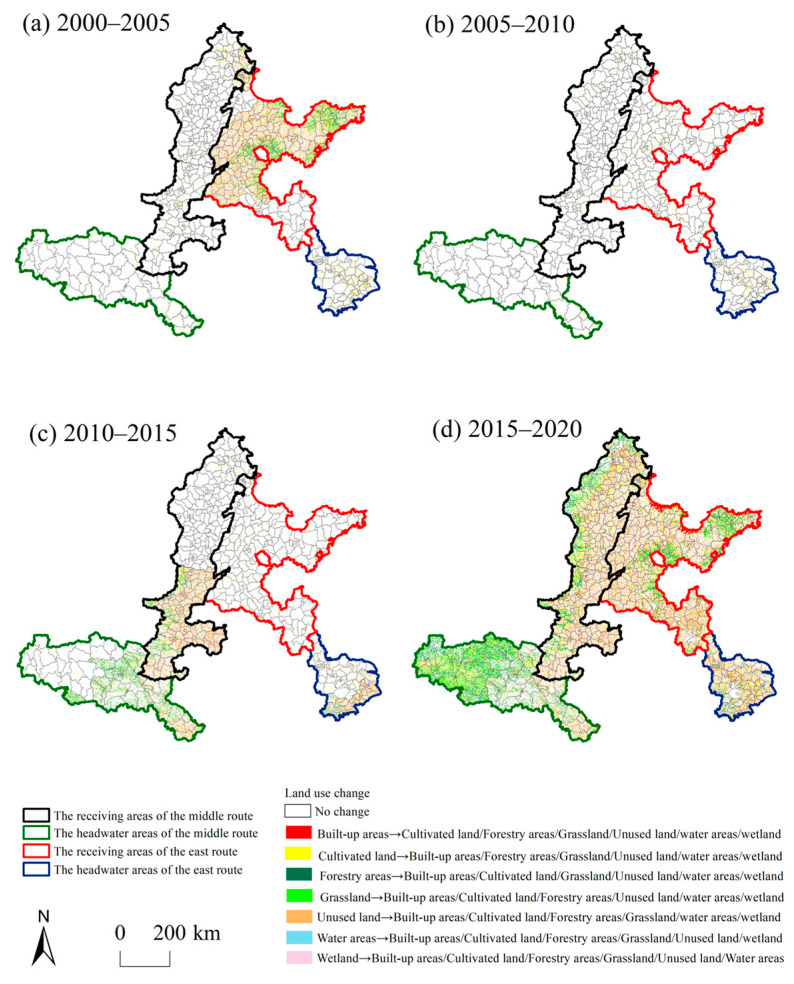
Land use transformation pattern in the study area during 2000–2005, 2005–2010, 2010–2015, and 2015–2020. (**a**) Land use transformation pattern in the study area from 2000 to 2005, (**b**) Land use transformation pattern in the study area from 2005 to 2010, (**c**) Land use transformation pattern in the study area from 2010 to 2015, (**d**) Land use transformation pattern in the study area from 2015 to 2020.

**Figure 4 ijerph-20-05069-f004:**
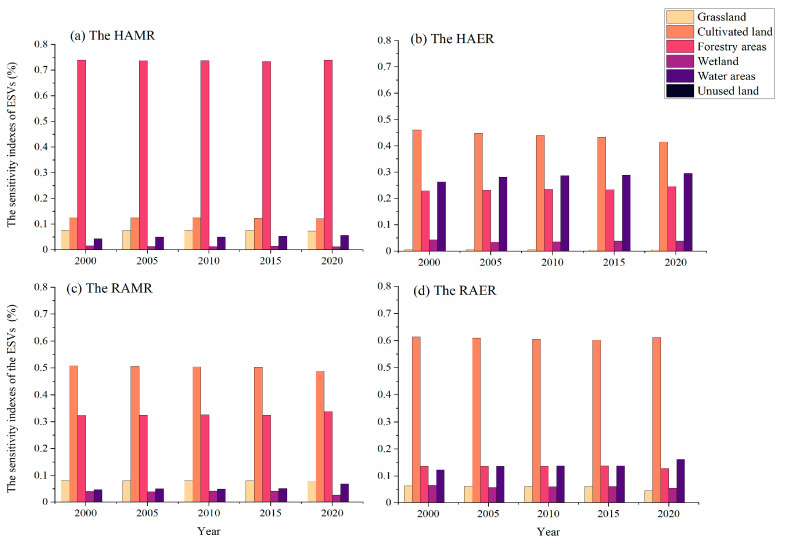
Sensitivity indexes in ecosystem service values of all land use types in the HAMR, HAER, RAMR, and RAER. (**a**) Sensitivity indexes of ecosystem service values in the HAMR, (**b**) Sensitivity indexes of ecosystem service values in the HAER, (**c**) Sensitivity indexes of ecosystem service values in the RAMR, (**d**) Sensitivity indexes of ecosystem service values in the RAER.

**Figure 5 ijerph-20-05069-f005:**
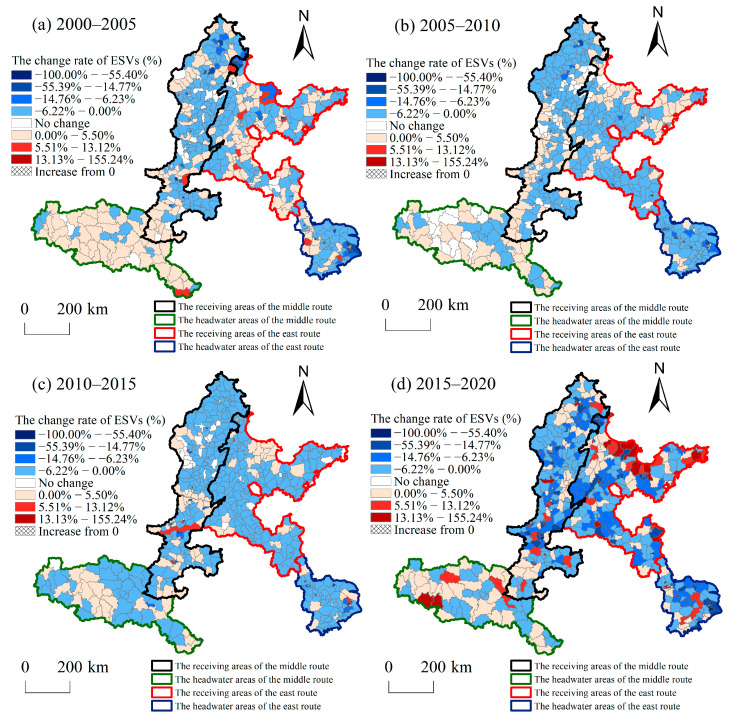
Change rates patterns of ecosystem service value in the study area during 2000–2005, 2005–2010, 2010–2015, and 2015–2020. (**a**) Change rates patterns of ecosystem service value in the study area from 2000 to 2005, (**b**) Change rates patterns of ecosystem service value in the study area from 2005 to 2010, (**c**) Change rates patterns of ecosystem service value in the study area from 2010 to 2015, (**d**) Change rates patterns of ecosystem service value in the study area from 2015 to 2020.

**Figure 6 ijerph-20-05069-f006:**
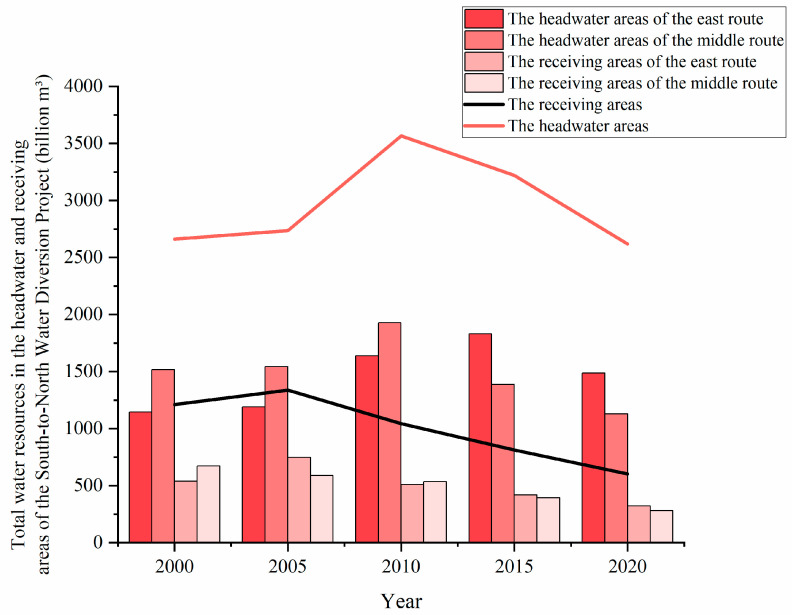
Total water resources of different subareas in five years.

**Table 1 ijerph-20-05069-t001:** Economic value of grain provided per unit area of each province/municipality directly under the central government (yuan/m^2^).

Region	Beijing	Tianjin	Hebei	Shanghai	Jiangsu	Zhejiang	Shandong	Henan	Hubei	Chongqing	Shaanxi
The economic value per unit area	0.043	0.045	0.036	0.030	0.046	0.050	0.050	0.052	0.050	0.027	0.018

**Table 2 ijerph-20-05069-t002:** ESVs of all land use types in the HAMR from 2000 to 2020 (million yuan).

Type	Grassland	Cultivated Land	Forestry Areas	Wetland	Water Areas	Unused Land	Total
2000	7850.93	13,077.70	76,668.62	1615.68	4450.45	2.02	103,665.41
2005	7876.86	12,936.56	76,693.81	1350.77	5148.43	2.02	104,008.46
2010	7876.84	12,892.52	76,666.47	1430.31	5215.74	2.02	104,083.90
2015	7872.68	12,772.65	76,213.17	1524.03	5493.00	2.24	103,877.76
2020	7590.36	12,620.25	77,290.06	1181.49	5851.80	2.16	104,536.12
2000–2005	25.93	−141.15	25.20	−264.91	697.98	0	343.05
2005–2010	−0.02	−44.04	−27.35	79.54	67.31	0	75.44
2010–2015	−4.16	−119.87	−453.30	93.72	277.26	0.21	−206.14
2015–2020	−282.33	−152.40	1076.90	−342.54	358.81	−0.08	658.36
2000–2020	−260.57	−457.45	621.45	−434.19	1401.35	0.14	870.71

**Table 3 ijerph-20-05069-t003:** ESVs of all land use types in the HAER from 2000 to 2020 (million yuan).

Type	Grassland	Cultivated Land	Forestry Areas	Wetland	Water Areas	Unused Land	Total
2000	133.98	14,548.46	7256.22	1350.90	8322.02	0.94	31,612.51
2005	131.40	14,019.05	7250.11	1098.62	8824.01	1.07	31,324.26
2010	130.62	13,564.63	7245.27	1104.11	8863.91	1.00	30,909.54
2015	153.04	13,197.00	7124.97	1158.94	8817.29	0.91	30,452.14
2020	147.65	12,211.38	7234.07	1126.52	8671.46	3.63	29,394.71
2000–2005	−2.58	−529.40	−6.11	−252.28	501.99	0.13	−288.26
2005–2010	−0.78	−454.43	−4.83	5.49	39.90	−0.07	−414.72
2010–2015	22.42	−367.63	−120.30	54.83	−46.62	−0.09	−457.39
2015–2020	−5.39	−985.62	109.10	−32.42	−145.83	2.72	−1057.43
2000–2020	13.67	−2337.08	−22.15	−224.38	349.44	2.69	−2217.80

**Table 4 ijerph-20-05069-t004:** ESVs of all land use types in the RAMR from 2000 to 2020 (million yuan).

Type	Grassland	Cultivated Land	Forestry Areas	Wetland	Water Areas	Unused Land	Total
2000	7559.31	47,881.69	30,472.08	3835.83	4438.59	4.11	94,191.60
2005	7533.45	47,471.46	30,455.31	3764.00	4669.98	2.91	93,897.11
2010	7508.15	47,288.55	30,464.16	3880.45	4590.50	2.94	93,734.75
2015	7509.79	47,014.94	30,382.31	3928.20	4772.70	3.06	93,610.99
2020	7283.87	44,660.90	30,843.06	2419.46	6238.02	1.31	91,446.62
2000–2005	−25.86	−410.23	−16.77	−71.83	231.40	−1.20	−294.49
2005–2010	−25.30	−182.91	8.85	116.45	−79.49	0.03	−162.36
2010–2015	1.64	−273.62	−81.85	47.75	182.20	0.12	−123.76
2015–2020	−225.93	−2354.04	460.75	−1508.74	1465.32	−1.75	−2164.38
2000–2020	−275.44	−3220.80	370.98	−1416.36	1799.44	−2.80	−2744.98

**Table 5 ijerph-20-05069-t005:** ESVs of all land use types in the RAER from 2000 to 2020 (million yuan).

Type	Grassland	Cultivated Land	Forestry Areas	Wetland	Water Areas	Unused Land	Total
2000	5998.67	58,407.88	12,963.31	6192.02	11,648.09	87.45	95,297.42
2005	5768.51	57,776.81	12,903.06	5495.50	12,901.90	77.48	94,923.26
2010	5748.78	57,248.88	12,907.52	5688.91	12,989.27	75.27	94,658.63
2015	5752.91	56,837.75	12,898.97	5725.95	13,017.33	75.22	94,308.13
2020	4123.02	56,317.97	11,761.98	5019.76	14,899.99	14.38	92,137.11
2000–2005	−230.17	−631.06	−60.25	−696.53	1253.81	−9.97	−374.16
2005–2010	−19.72	−527.93	4.46	193.41	87.37	−2.21	−264.63
2010–2015	4.13	−411.13	−8.55	37.04	28.06	−0.06	−350.50
2015–2020	−1629.89	−519.78	−1136.99	−706.19	1882.67	−60.83	−2171.02
2000–2020	−1875.65	−2089.90	−1201.33	−1172.26	3251.90	−73.07	−3160.31

## Data Availability

The data that support the findings of this study are available from the corresponding author upon reasonable request.

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
