# Peer review of "Impacts of Land-Use Change on Ecosystem Services Value in the South-to-North Water Diversion Project, China"

_ijerph, 2023, doi:10.3390/ijerph20065069_

Round 1
Reviewer 1 Report
The paper used the land-use dynamic degree index and land-use transfer matrix to explore the dynamics of LUCs in the headwater and receiving areas. However, the novelty seems to be limited. Significant improvements (major revisions) are needed:
1. A deep literature review should be given, Land use change plays an important role in ecosystem services value. The driving factors of land use change should be mentioned. e.g. 10.1080/15481603.2018.1507074; 10.1016/j.jag.2021.102475; 10.3390/ijerph110303215.
2. In section 3.2, ” land-use data (2000, 2005, 2010, 2015, and 2020) were generated at a 1 km resolution by manual visual interpretation”. The interpretation accuracies of land-use data should be mentioned.
3. The “results” part is boring. Take section 4.1.1 to section 4.1.4 as an example, the table and results analysis are similar and there is no focus. It is suggested to make a further summary of this part and use a more intuitive chart.
4. There are many typing mistakes in equations. Please check them carefully.
5. According to the legend in Figure 2, it is difficult to distinguish the map attributes.
Reviewer 2 Report
General comments:
In this manuscript, authors take the area where the South-to-North Water Diversion Project flows through as the research area to explore the impact of land use in this area on ecosystem services in the headwater areas and the receiving areas, so as to protect the surrounding ecological environment. The topic of this manuscript follows the international research hotspot. However, several questions should be carefully addressed, and the manuscript can be accepted after Major revisions. Please see below for the comments, which may be helpful for improving the quality of the manuscript.
Abstract
Line 25: What does the "wider" mean? Is there any other noun to explain.
Study area
1. Line 118: The numbers of "3" should be uploaded.
2. Line 121-122: How this conclusion was obtained, please attach the references.
3. Figure 1: 1) The China border in your picture is close to the outer frame, so I think the China map above can leave a space on the left. 2) Some of the provincial captial annotation in the second figure is blurred. I hope to adjust the color of the bottom line or annotate in the space with an arrow. 3) I think the third graph legend line element should be at the top.
Materials and Methods
1. Full-text formula labels are not aligned.
2. Line 136: The formula is wrong.
3. Line 138-141: 1) "An" is in an incorrect format. 2) "U" in the formula (2) does not match "u" in the explanation below. 3) The article doesn't explain what "i" and "j" represent respectively, so I don't understand equation (2).
4. Line 146: Whether the last column in the second row of the formula should be "S2n", and there is a punctuation mark before the "j".
5. Line 155-157: Please double check the character formatting in the explanation of the formula.
6. Line 170-172: Same question as above.
7. Line 185-188: Why are the letters subscripted in the formula but not in this paragraph, please double check, and the "i" is not in the right format
8. Line 200-201: Where did you get the 50% result from?
9. Line 208: My suggestion is put the ”Data sources“ in front of the " Research methods".
Results
1. Line 265-266: This study shows the headwater areas first and then the receiving areas in the results, and this sentence is based on the comparison of the results of the later studies, so I think this sentence can be placed in the conclusion and should not be in this paragraph.
2. Line 271-285: There is some overlap between the results of the 2000-2005 and 2005-2010 time periods, is it possible to combine the results of the two time periods for analysis? The same issue is also true for the time periods 2010-2015 and 2015-2020.
3. Line 294-295: In Table 4, the rate of change of cultivated land is higher than that of grassland and forestry areas, so how can it be the lowest rate of change type?
4. Line 299: "gradually increased"? However, Table 4 shows a decrease in 2005-2010 compared to 2000-2005.
5. Line 307: No transfer of the same land use type.
6. Line 318-319: There is an ambiguity in this sentence. Please explain clearly whether both cultivated land and built-up areas is converted mainly to grassland and forestry land or one of them is converted mainly to grassland and forestry land.
7. Line 328: What does this statement mean and how is the value of 0.750% obtained?
8. Line 340-346: The types of land use transferred in and out of these two time periods are consistent and I believe they can be combined for analysis.
9. Line 348-350: This analysis is ambiguous and I don't understand it very well. Are both land use types, cultivated land and build-up areas, converted mainly to grassland and forestry land, or is cultivated mainly converted to grassland and build-up areas mainly converted to forestry land, please explain.
10. Figure 2: Are the land use type change legends the same for all four time periods, if they are the same I think they can be shown only once and the image layout adjusted.
11. Line 369-371: Firstly, why is forestry areas the largest in 2000-2020, the water areas value in the table is even larger than forestry areas, secondly, how did you get 73%, is it the ratio of forestry areas to total, why did I calculate 71.4%, it doesn't exceed 73%.
12. Line 371-373: How did you get the 45.45% value and does the percentage here refer to the area or something else, please explain clearly. I don't really understand the sorting at the end either.
13. Line 376-377: Forestry areas and unused land values are positive for 2000-2020, why not an increase.
14. Line 431: Wetland appears twice in this sentence.
15. Line 483-485: How did the result come from? I can't observe these conclusions from Figure 3. Do these two sentences mean the same thing? Please explain carefully.
16. Line 495: Does the "above-14.77%" range mean "-100.00% - -14.76%" or "-14.76% - 155.24%"?
17. Line 496-497: The analysis shows that elasticity indexes of ESV in the range of -14.76%-0% are common throughout the study area, so what is the use of this "mainly distributed"?
18. Figure 4: Adjustment of the legend colour, the partition borders in the drawing are not clear.
Discussion
1. Line 552: This title is the impact of the policy, but in the discussion below there is no mention of the impact of the policy on the outcome and no specific measures are listed in the recommendations made.
2. Line 559: How to protect, please enumerate specific recommendations and measures.
Conclusions
Line 585-586: What does the change here represent, land use type area change or something else.
Reviewer 3 Report
The paper presents the impacts of land use change affecting ecosystem services functions along the South-to-North Water Diversion Project in China.
In my opinion, the article has some weaknesses to be addressed before its publication.
• You have to better clarify the terms used as the basis of your research such as ESV, and LUC in the Introduction section.
• What gap in the previous literature does the paper cover? You have to highlight this by making a literature review.
• What is the source of data used for applying your methodology approach?
• You do not discuss the results in light of previous research.
• What are your suggestions for policy-makers according to your conclusions?
Round 2
Reviewer 1 Report
All my comments have been well responsed. I have no more question.
Reviewer 2 Report
Thank you for your response. The article has been revised and is acceptable for publication.
Reviewer 3 Report
The authors improved the manuscript according to the reviewers' comments.